# Can Vitamin D Reduce Glucocorticoid-Induced Adverse Effects in Patients with Giant Cell Arteritis? Results from 1568 Patients in the Spanish ARTESER Registry [note 1]

**DOI:** 10.3390/nu17203291

**Published:** 2025-10-20

**Authors:** Gastón A. Ghio, Marta Domínguez-Álvaro, Iñigo Hernández Rodríguez, Elisa Fernández-Fernández, Maite Silva-Díaz, Joaquín M. Belzunegui, Clara Moriano, Julio Sánchez Martín, Javier Narváez, Eva Galíndez Agirregoikoa, Anne Riveros Frutos, Francisco Ortiz Sanjuán, Tarek C. Salman Monte, Margarida Vasques Rocha, Carlota L. Iñiguez, Alicia García Dorta, Clara Molina Almela, María Alcalde Villar, José L. Hernández, Santos Castañeda, Ricardo Blanco

**Affiliations:** 1Rheumatology Department, Hospital Universitari Mùtua de Terrassa, 08221 Barcelona, Spain; 2Research Unit, Sociedad Española de Reumatología, 28001 Madrid, Spain; 3Rheumatology Department, Complejo Hospitalario Universitario de Vigo, 36312 Galicia, Spain; 4Rheumatology Department, Hospital Universitario La Paz, 28046 Madrid, Spain; 5Rheumatology Department, Complejo Hospitalario Universitario de A Coruña, 15006 La Coruna, Spain; 6Rheumatology Department, Hospital Universitario Donosti, 20014 Donostia, Spain; 7Rheumatology Department, Hospital Universitario de León, 24071 León, Spain; 8Rheumatology Department, IDIVAL Immunopathology Group, Hospital Universitario Marqués de Valdecilla, 39008 Santander, Spain; 9Rheumatology Department, Hospital Universitario Bellvitge, Hospitalet de Llobregat, 08907 Barcelona, Spain; 10Rheumatology Department, Hospital Universitario de Basurto, 48013 Bilbao, Spain; 11Rheumatology Department, Hospital Germans Trias i Pujol, 08916 Badalona, Spain; 12Rheumatology Department, Hospital Universitario La Fe, 46026 València, Spain; 13Rheumatology Department, Hospital del Mar, 08003 Barcelona, Spain; 14Rheumatology Department, Hospital Universitario Araba, 01009 Vitoria-Gasteiz, Spain; 15Rheumatology Department, Hospital Universitario Lucus Augusti, 27003 Lugo, Spain; 16Rheumatology Department, Hospital Universitario de Canarias, 38320 La Laguna, Spain; 17Rheumatology Department, Consorci Hospital General Universitari de Valencia, 46014 València, Spain; 18Rheumatology Department, Hospital Universitario Severo Ochoa Leganés, 28914 Leganés, Spain; malcaldevillar@hotmail.es; 19Internal Medicine Department, Hospital Universitario Marqués de Valdecilla, 39008 Santander, Spain; hernandezjluis@gmail.com; 20IDIVAL, Department of Medicine and Psychiatry, University of Cantabria, 39005 Santander, Spain; 21Rheumatology Department, IIS-Princesa, Hospital Universitario de La Princesa, Universidad Autónoma de Madrid, 28049 Madrid, Spain; scastas@gmail.com

**Keywords:** giant cell arteritis, glucocorticoids, vitamin D, supplementation, adverse drug reactions

## Abstract

Objective: To determine whether oral vitamin D supplementation reduces the risk of glucocorticoid (GC)-associated severe adverse events (SAEs) in patients with giant cell arteritis (GCA) included in the Spanish ARTESER registry. Methods: The ARTESER registry collected data from patients diagnosed with GCA across 26 Spanish public hospitals between June 2013 and March 2019. SAEs were defined as fatal, life-threatening, or requiring hospitalization. Patients were categorized according to the use or non-use of oral vitamin D supplements. Incidence rates (IRs) of SAEs were expressed per person-year with 95% confidence intervals (CIs). Cox proportional hazards models assessed vitamin D supplementation and its interaction with cumulative glucocorticoid dose. Results: Of 1568 patients (mean age 76.9 ± 8.1 years; 70.1% women) receiving GC, 120 (7.6%) experienced SAEs (IR 0.039; 95% CI 0.033–0.047). Vitamin D supplementation was documented in 1186 (75.6%) compared with 382 (24.4%) non-supplemented patients. SAE incidence was similar in supplemented (n = 89; 7.5%; IR 0.038, 95% CI 0.030–0.046) and non-supplemented patients (n = 31; 8.1%; IR 0.045, 95% CI 0.031–0.064) (*p* = 0.387). Multivariable Cox regression showed a significant interaction between vitamin D supplementation and cumulative glucocorticoid dose (interaction term HR 0.90; *p* = 0.033), consistent with a dose-dependent protective effect. Conclusions: Vitamin D supplementation was not independently associated with a lower incidence of GC-related SAEs, likely due to residual confounding factors. However, the interaction with cumulative GC exposure suggests a modulatory effect. Prospective studies incorporating stratified baseline vitamin D assessments are warranted.

## 1. Introduction

Glucocorticoids (GCs) have been the mainstay of therapy for many systemic autoimmune disorders since their introduction [1]. Their toxicity—often serious and occasionally fatal—is closely related to cumulative exposure (dose and duration) and remains the principal barrier to long-term treatment [2,3,4].

Among systemic vasculitides, giant cell arteritis (GCA) is particularly responsive to GCs because of their potent immunosuppressive activity; therefore, GCs form the cornerstone of current management strategies [5,6,7]. For example, initial prednisone doses in GCA often range from 40 to 60 mg/day, with tapering frequently extending beyond 12–18 months. Relapse rates approach 40–50% in most series, contributing to cumulative toxicity. Reported incidences of major GC-related complications include 30–50% for osteoporosis and fractures, 20–30% for diabetes, and 15–25% for serious infections [8,9,10,11].

GCs disrupt several physiological systems, particularly calcium–phosphate regulation and bone remodeling, and are linked to ocular complications (cataracts, glaucoma), psychiatric manifestations (psychosis, depression), and exacerbation of pre-existing mental illness [1]. Their metabolic and vascular effects—including insulin resistance and diabetes, hypertension, dyslipidemia, and reduced fibrinolysis—promote atherosclerosis and coronary disease, contributing to increased morbidity and mortality [4]. In parallel, their immunosuppressive effects increase susceptibility to infections [9].

Vitamin D, a steroid hormone synthesized mainly in the skin upon ultraviolet-B exposure, exerts its biological actions through the vitamin D receptor, a nuclear transcription factor regulating numerous genes [12]. Its primary role is maintaining calcium–phosphate homeostasis and skeletal metabolism; however, low circulating levels have been associated with increased risks of cardiovascular disease, autoimmunity, infections, osteoporotic fractures, and all-cause mortality [13].

Beyond skeletal effects, vitamin D modulates innate immune responses, reduces T-cell co-stimulation, and lowers pro-inflammatory cytokine production [14,15,16]. These mechanisms have raised interest in the potential role of vitamin D as an adjunctive therapy in autoimmune conditions. Additional reported properties include synergistic effects with certain antibiotics and antioxidant activity [17]. Several studies also suggest a role in blood pressure regulation [18,19,20,21].

Emerging data in polymyalgia rheumatica suggest that vitamin D status may be associated with the achievement of remission [22], which is biologically plausible given its immunomodulatory effects [23,24]; however, such findings should not be directly extrapolated to giant cell arteritis.

Glucocorticoids and vitamin D share overlapping pathways in both bone and immune regulation. Low vitamin D levels have been associated with glucocorticoid use [13], and supplementation with vitamin D3 has been shown to protect against prednisolone-induced bone loss [25]. These observations provide the rationale to investigate whether vitamin D supplementation may attenuate glucocorticoid-related toxicity in giant cell arteritis.

Against this background, we aimed to assess whether vitamin D exposure is associated with a lower risk of glucocorticoid-related severe adverse events (SAEs) in patients with GCA included in a multicenter Spanish registry. As a secondary objective, we explored its potential effects on selected neuropsychiatric AEs, with particular focus on depression.

Preliminary data from this cohort were presented as an abstract at ACR Convergence 2024 [26]. This manuscript provides an expanded and updated analysis of that work.

## 2. Materials and Methods

### 2.1. Study Design

ARTESER (ARTEritis, Sociedad Española de Reumatología [SER]) is a nationwide, multicenter, longitudinal observational registry sponsored by the Spanish Society of Rheumatology [27]. For this analysis, we conducted a retrospective review of clinical data from patients diagnosed with giant cell arteritis (GCA) between 1 June 2013 and 29 March 2019, across 26 Spanish referral centers.

### 2.2. Patient Selection and Recruitment

Eligible patients were aged ≥50 years, consistent with the epidemiology of GCA and the age threshold specified in the 1990 ACR classification criteria [28], and met at least one of the following criteria: (i) a positive diagnostic test—temporal artery biopsy (TAB), ultrasound (US), or another relevant imaging modality; (ii) fulfillment of ≥3 of the 5 1990 American College of Rheumatology (ACR) classification criteria for GCA; or (iii) a clinical diagnosis of GCA by the treating physician in accordance with recent recommendations.

Of the 1675 individuals enrolled in ARTESER, 1568 who had received glucocorticoids were included in this study. Patients were stratified by oral vitamin D use into a supplemented cohort (n = 1186) and a non-supplemented cohort (n = 382) (Figure 1).

### 2.3. Severe Adverse Events

All severe adverse events (SAEs) arising during glucocorticoid therapy were systematically recorded according to the ARTESER registry protocol. SAEs were defined as events that were fatal or life-threatening, required prolonged hospitalization, resulted in persistent or significant disability, were associated with congenital anomalies, or were considered clinically significant by the attending physician.

### 2.4. Variables and Measurements

Data were extracted from medical records, including sociodemographic characteristics, comorbidities, and laboratory parameters at diagnosis. Treatments and adverse events were also recorded. For systemic glucocorticoid therapy, detailed information was collected on regimen, route of administration, and cumulative dose. To improve accuracy and minimize computational errors, cumulative glucocorticoid exposure was calculated using a purpose-built tool developed for this registry, the CortiSER Calculator [29].

### 2.5. Statistical Analysis

Continuous variables are presented as mean ± standard deviation (SD), and categorical variables as frequencies and percentages. Baseline characteristics were compared between vitamin D-supplemented and non-supplemented patients using bivariate analyses. For each group, incidence rates (IRs) of SAEs with 95% confidence intervals (CIs) were estimated. To evaluate whether vitamin D was associated with a reduced risk of glucocorticoid-related SAEs, we fitted a multivariable Cox proportional hazards model. Variables with *p* < 0.15 in bivariate analyses and those deemed clinically relevant were included. Statistical significance was set at *p* < 0.05. All analyses were performed with IBM SPSS Statistics, Version 28.0 (IBM Corp., Armonk, NY, USA).

### 2.6. Ethical Considerations

The study was approved by the Medical Research Ethics Committee of the Hospital Universitario Marqués de Valdecilla, Santander, Spain (Number code: 05/2019) and conducted in accordance with the Declaration of Helsinki. Data processing complied with the General Data Protection Regulation (GDPR), specifically Regulation (EU) 2016/679 of the European Parliament and of the Council of 27 April 2016 on the protection of natural persons regarding the processing and free movement of personal data.

## 3. Results

### 3.1. Sociodemographic Characteristics

We analyzed 1568 patients treated with oral and/or parenteral glucocorticoids. Of these, 468 (29.8%) were men and 1100 (70.2%) were women. The mean age at diagnosis was 76.9 ± 8.1 years.

Vitamin D supplementation had been prescribed in 1186/1568 (75.6%) patients (Figure 1). At diagnosis, supplemented patients were slightly younger than non-supplemented patients (76.7 ± 8.0 vs. 77.7 ± 8.2 years; *p* = 0.017) (Table 1).

A history of cardiovascular disease was documented in 345/1568 (22.0%) patients: 110/382 (28.8%) in the non-supplemented group and 235/1186 (19.8%) in the supplemented group (Table 1). Prior osteoporosis was recorded in 265/1568 (16.9%) patients, including 230/1186 (19.4%) in the supplemented group and 35/382 (9.2%) in the non-supplemented group (Table 1).

### 3.2. Incidence of Severe Adverse Events

Among the 1568 patients treated with glucocorticoids from initiation to last follow-up, 120 SAEs were recorded, corresponding to an overall (cumulative) incidence rate of 0.039 (95% CI, 0.033–0.047). When stratified by vitamin D supplementation, 89/1186 (7.5%) supplemented patients experienced SAEs (incidence rate 0.038, 95% CI, 0.030–0.046), compared with 31/382 (8.1%) non-supplemented patients (incidence rate 0.045, 95% CI, 0.031–0.064). The between-group difference was not statistically significant (*p* = 0.387) (Table 2).

In the psychiatric subgroup, the incidence rate of psychiatric SAEs—specifically depression—was 0.007 events per person-year (95% CI, 0.005–0.011). Among patients receiving vitamin D, the incidence rate was 0.009 (95% CI, 0.006–0.014) compared with 0.0014 (95% CI, 0.000–0.008) in those without supplementation; this difference was statistically significant (*p* = 0.022) (Table 2).

### 3.3. Protective Role of Vitamin D Assessed by Cox Regression Analysis

A multivariable Cox proportional hazards model was fitted to evaluate the association between vitamin D supplementation and glucocorticoid-associated SAEs. Hypertension, prior cardiovascular disease, and vitamin D use emerged as independent variables significantly associated with SAE occurrence (*p* < 0.05).

A statistically significant interaction between cumulative glucocorticoid dose and vitamin D supplementation was identified (interaction term HR = 0.90, *p* = 0.033), consistent with a protective modification of risk in supplemented patients. Taken together, these findings suggest that concomitant vitamin D may attenuate SAE risk during glucocorticoid therapy (Figure 2).

## 4. Discussion

In this large, multicenter GCA cohort, we evaluated whether vitamin D supplementation modifies the risk of glucocorticoid-related SAEs. This question is highly relevant in GCA, where prolonged glucocorticoid exposure is common and patients are typically older, frequently multimorbid, and more vulnerable to treatment-related toxicity. Although supplementation did not significantly lower the overall incidence of SAEs, the multivariable analysis identified a clinically relevant interaction with cumulative glucocorticoid exposure: as dose burden increased, patients receiving vitamin D showed an attenuated risk of SAEs (Figure 2). This pattern supports a modulatory rather than uniformly preventive effect of vitamin D on glucocorticoid toxicity. In other words, any benefit is unlikely to be uniform across all patients or outcomes but may emerge under conditions of higher steroid burden, consistent with a context-dependent effect.

Long-term glucocorticoid therapy is well known to entail numerous adverse effects. Yet, specifically in GCA, evidence on adjunctive strategies to mitigate steroid toxicity remains scarce and heterogeneous, with most data derived from observational cohorts rather than randomized evaluations [6]. As in most registry-based series, our cohort comprised predominantly women and older adults, with expected comorbidities such as cardiovascular disease and osteoporosis [27,30]. In routine practice, given advanced age and glucocorticoid exposure, vitamin D is frequently prescribed to preserve bone health and prevent fractures in GCA [31], a practice aligned with national and international guidance on glucocorticoid-induced osteoporosis. This guideline-driven prescribing pattern likely contributed to the high uptake of vitamin D in our cohort and may also introduce indication bias toward patients perceived to be at higher baseline risk [31,32]. Consistent with this, 75.6% of glucocorticoid-treated patients in ARTESER received vitamin D.

The burden of glucocorticoid-attributed AEs in our study was high and comparable to prior reports [1,7]. The spectrum of SAEs included clinically relevant complications expected under chronic steroid exposure, such as severe infections, osteoporotic fractures, and major cardiovascular events, all routinely captured within ARTESER. We focused on severe events because, due to the retrospective design, mild or moderate AEs may be undercaptured in electronic records. Despite the ample sample size, we did not demonstrate a reduction in overall SAE incidence with supplementation; however, the observed interaction with cumulative dose suggests that vitamin D may mitigate dose-dependent risk. This nuance is coherent with current understanding of glucocorticoid toxicity and with a plausible, context-dependent role for vitamin D.

Interpretation must consider potential residual confounding and indication bias inherent to real-world data. Additional challenges in registry studies include time-varying exposure to steroids and supplementation, potential misclassification of exposure, and incomplete capture of outpatient adverse events. Clinicians may preferentially supplement patients perceived to be at higher baseline risk (e.g., those with osteoporosis or higher cumulative glucocorticoid exposure), which could obscure benefits. Moreover, baseline serum vitamin D status, dosing regimens, and adherence—determinants of biological effect—were not systematically recorded. A threshold phenomenon is also conceivable, where only patients with marked deficiency derive measurable benefit. Furthermore, the absence of baseline 25-hydroxyvitamin D measurements and adherence data may dilute true effects by grouping biologically dissimilar patients into the same exposure category. It remains possible that vitamin D does not exert a strong main effect on SAE risk but modifies toxicity in the setting of greater glucocorticoid exposure.

The supplemented group showed a longer duration of symptoms prior to diagnosis; we consider it unlikely that this difference materially influenced the main findings (Table 1). Importantly, adjustment for major confounders and the observed interaction with cumulative steroid dose argue against a spurious association driven solely by baseline imbalances.

In the psychiatric subgroup (e.g., depression), patients receiving vitamin D supplementation exhibited a higher incidence of severe events. Although this appears discordant with prior observations linking vitamin D deficiency to neuropsychiatric conditions—particularly depression—the absolute incidence in our cohort was very low and warrants cautious interpretation. Sparse event counts and the possibility of differential detection or reporting across groups limit definitive conclusions regarding this signal, which should be explored in prospectively designed analyses [33]. Additionally, a sizable fraction of the general population has insufficient or deficient vitamin D levels without psychiatric symptoms [34].

Strengths of this study include the large, multicenter design, standardized SAE definitions within a national registry [27], and the use of a dedicated tool to estimate cumulative glucocorticoid exposure, which enhances measurement consistency across sites [29].

It is unlikely that vitamin D supplementation counteracts all GC-related adverse events. Rather, its potential benefits are likely context-dependent, with the strongest evidence in skeletal protection and possible contributions to cardiovascular risk modulation. Mechanistically, vitamin D enhances calcium–phosphate homeostasis, modulates bone remodeling, downregulates pro-inflammatory cytokines, and promotes immune tolerance, which together may attenuate specific toxicities of long-term glucocorticoid therapy

This study has limitations. First, its retrospective observational nature limits causal inference. Second, neither the specific formulation nor the dose of vitamin D was consistently recorded. Third, baseline serum vitamin D concentrations prior to glucocorticoid initiation were unavailable, precluding assessment of deficiency status or correction. Finally, reliance on SAEs may underestimate the frequency of less severe AEs; however, restricting outcomes to SAEs likely reduced variability and reporting bias. Despite these constraints, ARTESER represents one of the largest cohorts worldwide for evaluating GCA patients, improving the precision and external validity of our estimates.

From a practical standpoint, our findings support continued adherence to current bone-health recommendations for GCA patients receiving long-term steroids, while future work should clarify whether targeted vitamin D strategies benefit high-risk subgroups [31,32].

## 5. Conclusions

Vitamin D supplementation was not independently associated with a statistically significant reduction in the overall incidence of glucocorticoid-related SAEs. However, the observed interaction with cumulative glucocorticoid exposure (HR = 0.90; *p* = 0.033) suggests a modulatory effect that may attenuate dose-dependent toxicity. These findings highlight the importance of considering GC burden when evaluating potential protective factors. Future prospective studies, ideally stratified by baseline vitamin D status and incorporating biomarker-based monitoring, are warranted to clarify whether supplementation can meaningfully reduce glucocorticoid-related complications in GCA and other autoimmune conditions.

## Figures and Tables

**Figure 1 nutrients-17-03291-f001:**
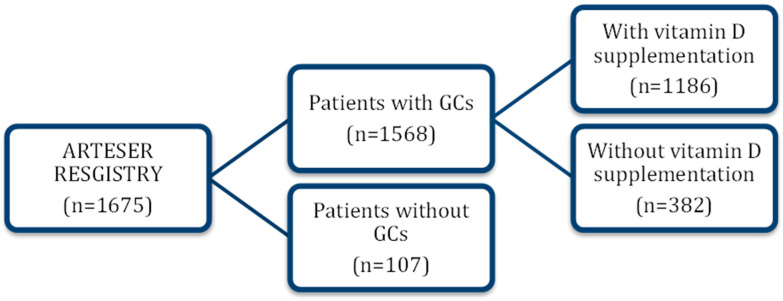
Patient flow diagram (ARTESER). Flow of patients with giant cell arteritis included in the ARTESER registry and selection of the analytic cohort for the present study (vitamin D exposed vs. non-exposed). GC: glucocorticoids.

**Figure 2 nutrients-17-03291-f002:**
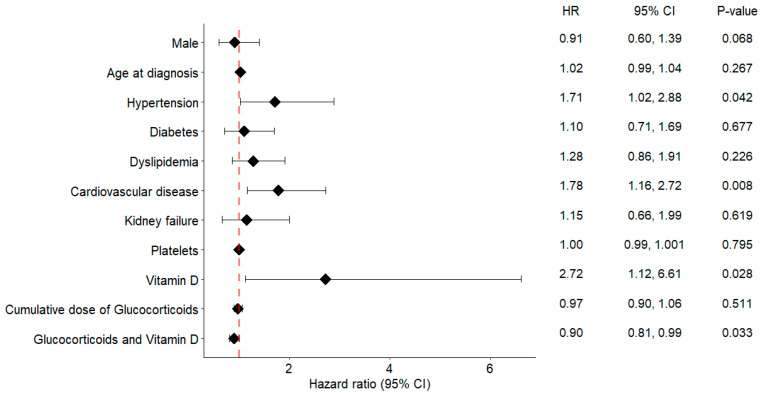
Adjusted associations between vitamin D supplementation and the primary composite outcome. Points indicate hazard ratios (HRs) and horizontal bars 95% confidence intervals (CIs) from multivariable Cox models adjusted for prespecified covariates (see Methods). The vertical dashed line marks the null effect (HR = 1.0). The row ‘Glucocorticoids and Vitamin D’ represents the interaction (product) term between cumulative glucocorticoid exposure and vitamin D supplementation. Abbreviations: GC, glucocorticoids; HR, hazard ratio; CI, confidence interval.

**Table 1 nutrients-17-03291-t001:** Baseline socio-demographic, clinical, and therapy-related characteristics of patients with giant cell arteritis included in the ARTESER registry.

	All Patients (n = 1568)	Not Taking Vitamin D (n = 382)	Taking Preventive Vitamin D (n = 1186)	*p*-Value
**Demographic data**				
Women, n (%)	1100 (70.2)	257 (67.3)	843 (71.1)	0.158
Age, mean (SD)	76.9 (8.1)	77.6 (8.2)	76.6 (8.0)	0.017
Duration of symptoms (months), mean (SD)	2.8 (5.5)	2.2 (3.2)	3.0 (6.0)	0.044
**Comorbidity**				
Smoking, n (%)	123 (8.3)	33 (9.0)	90 (8.1)	0.673
Previous CVD, n (%)	345 (22.0)	110 (28.8)	235 (19.8)	<0.001
Hypertension, n (%)	1014 (65.4)	260 (69.3)	754 (64.2)	0.067
Diabetes mellitus, n (%)	333 (21.7)	83 (22.6)	250 (21.4)	0.651
Dyslipidemia, n (%)	757 (49.2)	192 (51.9)	565 (48.3)	0.227
Osteoporosis, n (%)	265 (16.9)	35 (9.2)	230 (19.4)	<0.001
Obesity, n (%)	143 (16.3)	33 (16.3)	110 (16.2)	0.797
Chronic kidney disease, n (%)	158 (10.5)	51 (11.4)	117 (10.2)	0.527
**Type of treatment**				
Oral GC, n (%)	1163 (74.2)	306 (80.1)	857 (72.3)	0.001
Intravenous GCs, n (%)	3 (0.2)	2 (0.5)	1 (0.1)	0.001
Oral and intravenous GCs, n (%)	402 (25.7)	74 (19.4)	328 (27.7)	0.001
Cumulative oral dose of GCs, mean (SD)	7441.4 (4878.4)	6716.1 (4784.4)	7519.4 (4833.4)	<0.001

Abbreviations (in alphabetical order): CVD: cardiovascular disease; GCs: glucocorticoids; n: number; SD: standard deviation. Data are presented as mean ± SD, median (IQR), or n (%) as appropriate.

**Table 2 nutrients-17-03291-t002:** Proportion and incidence rate of severe adverse events and severe psychiatric adverse effects (depression) in patients with GCA undergoing glucocorticoid treatment with and without vitamin D supplementation.

	All Patients (n = 1568)	Patients Without Vitamin D (n = 382)	Patients with Vitamin D (n = 1186)	*p*-Value
Serious adverse events				
N (%)	120 (7.7)	31 (8.1)	89 (7.5)	
Incidence rate (95%CI)	0.039 (0.033–0.047)	0.045 (0.031–0.064)	0.038 (0.030–0.046)	0.387
Serious psychiatric adverse events				
N (%)	23 (1.5)	1 (0.3)	22 (1.9)	
Incidence rate (95%CI)	0.007 (0.005–0.011)	0.0014 (0.000–0.008)	0.009 (0.006–0.014)	0.022

Abbreviations: GCA: giant cell arteritis; CI: confidence interval; N: number.

## Data Availability

Data cannot be shared publicly due to patient privacy and ethical restrictions imposed by the ARTESER registry and the approving ethics committee.

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
