# Peer review of "Can Vitamin D Reduce Glucocorticoid-Induced Adverse Effects in Patients with Giant Cell Arteritis? Results from 1568 Patients in the Spanish ARTESER Registry"

_nutrients, 2025, doi:10.3390/nu17203291_

Round 1

Reviewer 1 Report

Comments and Suggestions for Authors

Thank you for the opportunity to review your manuscript titled "CAN VITAMIN D REDUCE GLUCOCORTICOID-INDUCED ADVERSE EFFECTS IN PATIENTS WITH GIANT CELL ARTERITIS? RESULTS FROM 1568 PATIENTS IN THE SPANISH ARTESER REGISTRY."

The article is interesting and the topic is clinically relevant, given the high burden of glucocorticoid toxicity and the widespread empirical use of vitamin D. The large sample size (n=1568) is a major strength, and the use of a standardized tool for glucocorticoid exposure is a strength.

I have some comments that, if addressed, can help improve the clarity and impact of the manuscript.

Major comments

The major limitation, which is already outlined in the manuscript, is the missing data regarding the type of vitamin D supplementation (i.e., Cholecalciferol vs. Calcitriol, weekly/daily dosage, units) and the baseline measured vitamin D status (25-OH vitamin D).

  • You have presented interesting results regarding psychiatric SAEs. I would like to see if it is possible to perform further subgroup analyses for other types of SAEs, such as infections, cardiovascular events, or bone fractures, to see if the protective effect of vitamin D varies across different outcomes.
  • I recommend using Kaplan-Meier survival curves to visually represent the cumulative incidence of SAEs over time for patients who received vitamin D versus those who did not. It would be valuable to assess, if feasible, whether to further stratify these curves by glucocorticoid dose (e.g., high vs. low cumulative dose) to illustrate the interaction you found.

Minor comments

  • In the abstract, the sentence “Multivariable Cox regression showed a significant interaction 111 between vitamin D use and GC (hazard ratio 0.90; p=0.033), indicating a dose-dependent 112 protective effect of vitamin D supplements” is not clear. Please clarify if you meant GC dosage instead of GC, as this distinction is critical to the interpretation of your main finding.
  • In the introduction, when you discuss the rationale for vitamin D (lines 144-148), you might also mention that in the closely related disease of PMR, the response to vitamin D supplementation has been recently associated with the achievement of remission (PMID: 40944227).

Author Response

Reviewer #1

The article is interesting and the topic is clinically relevant, given the high burden of glucocorticoid toxicity and the widespread empirical use of vitamin D. The large sample size (n=1568) is a major strength, and the use of a standardized tool for glucocorticoid exposure is a strength.

I have some comments that, if addressed, can help improve the clarity and impact of the manuscript.

Response: We sincerely thank the reviewer for his/her constructive assessment regarding clinical relevance, sample size, and the standardized approach to glucocorticoid (GC) exposure.

Major comments

  1. Comment: The major limitation, which is already outlined in the manuscript, is the missing data regarding the type of vitamin D supplementation (i.e., Cholecalciferol vs. Calcitriol, weekly/daily dosage, units) and the baseline measured vitamin D status (25-OH vitamin D).

    Response
    : We thank the reviewer for emphasizing this important point. ARTESER does not systematically capture vitamin D formulation, dosing, or baseline 25(OH)D due to the main objectives of this registry are related to knowledgement epidemiology, diagnosis and therapy of GCA in Spain. We have strengthened the Limitations section to acknowledge potential exposure misclassification and the impossibility of threshold analyses by deficiency status, and we highlight the need for prospective, biomarker-stratified studies.

Manuscript change: Discussion → Limitations paragraph expanded and revised accordingly. (Conclusions; see tracked file: page 9; paragraph 6)

  1. Comment:
    You have presented interesting results regarding psychiatric SAEs. I would like to see if it is possible to perform further subgroup analyses for other types of SAEs, such as infections, cardiovascular events, or bone fractures, to see if the protective effect of vitamin D varies across different outcomes.

Response: We appreciate this valuable suggestion. We assessed feasibility; however, sparse counts per SAE subtype would lead to unstable adjusted models and multiplicity concerns. To avoid over-interpretation, we refrain from additional underpowered models. Instead, we added a concise descriptive summary of SAE distribution and clarified our rationale.

Manuscript change: Results: Descriptive line added (page 6; Paragraph 5); Discussion: Justification added regarding power and multiplicity (page 8; paragraph 3).

  1. Comment: I recommend using Kaplan-Meier survival curves to visually represent the cumulative incidence of SAEs over time for patients who received vitamin D versus those who did not. It would be valuable to assess, if feasible, whether to further stratify these curves by glucocorticoid dose (e.g., high vs. low cumulative dose) to illustrate the interaction you found.

Response: We sincerely thank the reviewer for this valuable suggestion. Unfortunately, the dataset assembled for this revision does not include the individual time-to-event fields (event dates and censoring times) required to construct unadjusted Kaplan–Meier plots. Producing these figures would require a new data extraction and the corresponding institutional approvals, which falls outside the scope and timeline of the current round. Our primary time-to-event inference therefore relies on multivariable Cox models. If the Editorial Office considers KM plots essential, we can initiate the necessary steps to obtain the required fields and provide the figures in a subsequent stage.

Minor comments

  1. Comment: In the abstract, the sentence “Multivariable Cox regression showed a significant interaction between vitamin D use and GC (hazard ratio 0.90; p=0.033), indicating a dose-dependent protective effect of vitamin D supplements” is not clear. Please clarify if you meant GC dosage instead of GC, as this distinction is critical to the interpretation of your main finding.

Response: The reviewer is completely right. The interaction concerns to cumulative glucocorticoid dose, not “GC” in general. Accordingly, we have revised the Abstract to make this more explicit.

Manuscript change: (Abstract: page 3; paragraph 2)

  1. Comment: In the introduction, when you discuss the rationale for vitamin D (lines 144-148), you might also mention that in the closely related disease of PMR, the response to vitamin D supplementation has been recently associated with the achievement of remission (PMID: 40944227).

Response: We appreciate the pointer to this new reference (included with number 22). We added a brief sentence noting the emerging PMR evidence and the biological plausibility, while cautioning against direct extrapolation to GCA.

Manuscript change: Introduction: page 4; paragraph 3.

Reviewer 2 Report

Comments and Suggestions for Authors

The author analyzes patient of clinical data whether vitamin D is associated with a reduced risk of severe glucocorticoid-associated adverse events, the starting point is solid, but the content still has significant room for improvement. Recommendations for authors to prepare a major revision.

  1. Line 107: “occasionally fatal- are”, Please refine the English expression.
  2. Line 111-113: It is recommended to illustrate with specific case examples, including dosage details, recurrence rates, and the incidence of glucocorticoid-associated adverse events.
  3. Line 117-120: These two sentences express a viewpoint derived from multiple sources, with the author referring to“Patients immunosuppressive effect increases the risk of infections”? Optimize the phrasing of the language.
  4. Introduction: It is recommended not to cite references at the end of a paragraph or conclude a paragraph with a reference.
  5. The introduction lacks systematic organization and logical coherence; it is recommended to restructure it. How are glucocorticoids linked to vitamin D? Are there any current studies indicating this connection, and what is the current state of research?
  6. The author divided 1,568 who had received glucocorticoids into vitamin D supplementation and non-supplementation groups. Could the additional vitamin D intake in their daily diets influence the study's results? How did the author mitigate such factors?
  7. Why Choose Patients Aged 50 and over?

Glucocorticoid-associated adverse events What is the incidence rate across different age groups? This information is not reflected in the text.

If the age distribution of patients in this study included children, young adults, and the elderly, it would be more convincing.

  1. As is well known, glucocorticoids can trigger various adverse reactions. Does supplemental vitamin D counteract all adverse reactions, or does it only address a specific type? The author did not delve into the molecular biological mechanisms of vitamin D.
  2. Conclusions: This article fails to present substantive findings or offer constructive recommendations regarding glucocorticoid-associated adverse events.
  3. Line 320: Author Contributions: This section does not specify the authors' contributions. Please add this information.
  4. The reference format does not comply with the journal's requirements.
  5.  Recommendations for authors to prepare a major revision.

Author Response

Comment (Line 107): “occasionally fatal—are” requires refinement.

Response: We thank the reviewer for this suggestion. According to the recommendation, we have rephrased the sentence as follows:

“Glucocorticoid toxicity—often serious and occasionally fatal—is closely related to cumulative exposure (dose and duration) and remains the main barrier to long-term treatment.” (Introduction; see tracked file, page 3; paragraph 4).

Comment (Lines 111–113): Reviewer requested specific examples (doses, relapse rates, incidence of GC-AEs).

Response: The reviewer is right. Accordingly, we have expanded the paragraph in the Introduction to add representative figures from prior studies, illustrating typical prednisone starting doses (40–60 mg/day), prolonged tapering dosage (often >12 months), relapse rates (~40–50%), and incidences of major GC-associated complications (osteoporosis, diabetes, infections). This contextual detail is supported by several references [8–11]. (Introduction; see tracked file; page 3; paragraph 5).

Comment (Lines 117–120):  Optimize wording about immunosuppression and infection risk.

Response: We agree and thank the reviewer for this suggestion. Consequently, we have modified this sentence as follows:
“Their immunosuppressive effects increase susceptibility to infections.”
(Introduction; see tracked file; page 4; paragraph 1).

Comment (Introduction structure/references): Improve logical flow; avoid ending paragraphs with a reference; explain GC–vitamin D link and current evidence.

Response: We thank the reviewer for this helpful observation. In accordance, the Introduction has been reorganized to present: (1) GC use and toxicity in GCA; (2) vitamin D biology; (3) rationale for studying their interaction with current evidence. References were redistributed so paragraphs do not end with a citation alone. (Introduction; see tracked file: pages 3 and 4).

Comment (dietary vitamin D):
Could daily intake affect results? How mitigated?

Response: We appreciate and accept the reviewer´s observation as an unmeasured confounder. Dietary intake and sunlight exposure were not systematically collected in ARTESER registry; this limitation is now explicitly expressed (Discussion; see tracked file: page 9; paragraph 2).

Comment (≥50 years): Why only patients ≥50?

Response: GCA occurs almost exclusively in individuals aged ≥50 years and this threshold is embedded in classification criteria. We have clarified this rationale in the Methods section (see tracked file: page 4; paragraph 8).

Comment (incidence by age groups / inclusion of children-young adults):

Response: We thank the reviewer for raising this issue. Age-stratified analyses are limited by the narrow, elderly distribution typical of GCA; inclusion of children/young adults is not applicable to this disease. This is clarified in the text (Methods/Discussion; see tracked file: page 4; paragraph 8 and page 8; paragraph 2).

Comment (scope of vitamin D effect & mechanisms): Does it counteract all AEs? Expand mechanisms.

Response: We thank the reviewer for the opportunity to clarify this point. In fact, we have added that vitamin D is unlikely to prevent all GC-related AEs; and effects may be context-dependent (e.g., bone, possibly CV risk). In addition, we have included a concise mechanistic paragraph (immune modulation, cytokine reduction, bone metabolism). (Discussion; see tracked file: (page 9, paragraph 6 and page 10, paragraph1).

Comment (Conclusions):
Not sufficiently constructive.

Response: We thank the reviewer for this clear recommendation. We have revised the conclusions in order to emphasize the interaction with cumulative GC exposure and to recommend prospective, biomarker-stratified studies (Conclusions; see tracked file: page 10; paragraph 1).

Comment (Author Contributions): Missing.

Response: The reviewer is completely right. Therefore, we have added detailed Author Contributions (CRediT) (Back matter) in page 10.

Comment (Reference format):
Not compliant.

Response: The reviewer is right again. We have reformatted all references to Nutrients style.

Furthermore, language has been carefully polished with the assistance of a native English scientific translator.

Round 2

Reviewer 1 Report

Comments and Suggestions for Authors

The manuscript is significantly improved in my opinion! I don't have further comments and I congratulate the authors for this nice study